# Direct Consideration of Process History During Intensified Design of Experiments Planning Eases Interpretation of Mammalian Cell Culture Dynamics

**DOI:** 10.3390/bioengineering12030319

**Published:** 2025-03-19

**Authors:** Samuel Kienzle, Lisa Junghans, Stefan Wieschalka, Katharina Diem, Ralf Takors, Nicole Erika Radde, Marco Kunzelmann, Beate Presser, Verena Nold

**Affiliations:** 1Development Biologicals, Boehringer Ingelheim Pharma GmbH & Co. KG, Birkendorferstraße 65, 88397 Biberach an der Riß, Germany; 2Institute of Biochemical Engineering, University of Stuttgart, Allmandring 31, 70569 Stuttgart, Germany; 3Institute for Stochastics and Applications, University of Stuttgart, Wankelstr. 5, 70563 Stuttgart, Germany; 4Global Computational Biology and Data Science, Boehringer Ingelheim Pharma GmbH & Co. KG, Birkendorferstraße 65, 88397 Biberach an der Riß, Germany

**Keywords:** statistical design of experiment, quality by design, mammalian bioprocess dynamics, dynamic process optimization, shifted factor settings

## Abstract

Intra-experimental factor setting shifts in intensified design of experiments (iDoE) enhance understanding of bioproduction processes by capturing their dynamics and are thus essential to fulfill quality by design (QbD) ambitions. Determining the influence of process history on the cellular responses, often referred to as memory effect, is fundamental for accurate predictions. However, the current iDoE designs do not explicitly consider nor quantify the influence of process history. Therefore, we propose the one-factor-multiple-columns (OFMC)-format for iDoE planning. This format explicitly describes stage-dependent factor effects and potential memory effects as across-stage interactions (ASIs) during a bioprocess. To illustrate its utility, an OFMC-iDoE that considers the characteristic growth phases during a fed-batch process was planned. Data were analyzed using ordinary least squares (OLS) regression as previously described via stage-wise analysis of the time series and compared to direct modeling of end-of-process outcomes enabled by the OFMC-format. This article aims to provide the reader with a framework on how to plan and model iDoE data and highlights how the OFMC-format simplifies planning, and data acquisition, eases modeling and gives a straightforward quantification of potential memory effects. With the proposed OFMC-format, optimization of bioprocesses can leverage which factor settings are most beneficial in which state of the mammalian culture and thus elevate performance and quality to the next level.

## 1. Introduction

Conducting mammalian bioprocesses in the pharmaceutical industry aims to deliver medical products of the highest quality to patients in need. A thorough understanding and optimization of the processes for each product is needed to reliably and efficiently achieve this goal [1,2]. Within the quality by design (QbD) framework, design of experiments (DoE) has been extensively used to not only understand the individual contribution of an input factor setting to the process but to learn how multiple factors jointly affect the bioprocess and product quality [3,4]. Such interactions between factors can only be accurately modeled if information on the consequences of different combinations of factor settings are contained in the data. This can be achieved if the factors are varied together. What is often neglected during process optimization is the heterogeneous nature of production processes. In fed-batch processes, which are widely used in industry [5], cells undergo characteristic phases. These are typically divided into the lag, exponential growth, transition, and production phases [6] (p. 64). During each phase, cells are in different states [7]. These states likely lead to different optimal factor settings. However, classical process optimization is time-invariant. By neglecting the dynamic nature of bioprocesses, a large potential of temporally optimal factor settings remains unused. To obtain the required understanding of process dynamics, different combinations of factor settings at various time points need to be studied.

Intensified design of experiments (iDoE) is an extension of classical DoE that allows for studying temporal aspects of factor effects like the relevance of timing and exposure, by introducing intra-experimental shifts of factor setpoints. These scheduled setpoint changes divide the process into experimental stages that may or may not represent biologically meaningful phases. The biological feasibility of iDoE was shown for procaryotic [8] and eukaryotic cells during the growth and production phases [9,10]. Two modeling approaches have been suggested for data with scheduled setpoint changes. Hybrid modeling utilizes a semi-parametric approach combining mass balances with machine learning approaches such as neural networks [11,12,13]. Stage-wise (SW) ordinary least squares (OLS) fits separate models for each iDoE stage and model predictions are concatenated [9,10]. To obtain a prediction for the endpoint, the entire time series needs to be modeled, which is inconvenient. A further disadvantage is that time series require multiple sampling. For some responses, such as product quality, time series measurements and analytical processing can be very costly or practically infeasible. Additionally, extensive pre-processing of the data such as division into intervals, rebasing, and annotation with initial values might be necessary. In the typical time series format, a column representing the time is accompanied by one column for each input factor varying over time or across bioreactors (one-factor-one-column (OFOC)-format).

In theory, iDoE could be used to reduce the experimental burden by the number of factor setting combinations that are covered within one bioreactor [8,11,14]. However, the number of samples for the associated time series is not reduced. Practically, the multiple by which a compression of experimental burden can be reduced depends on the response kinetics of the system. Studies in perfusion systems have shown that Chinese hamster ovary (CHO) cells require 3–4 days to adapt to mild intra-experimental factor setpoint changes [15]. Thus, maximally a 4-fold reduction in a 14-day fed-batch process could be achieved. Moreover, potentially introduced memory effects need to be considered to not distort the estimation of which effect a factor has in a given stage [15]. Memory effects describe the influence that previous stages’ factor settings have on cellular response behavior during later stages of the process [11,15,16]. Illustrated on a classical fed-batch process, this means that preceding stages’ factor settings may lead to a shift in the characteristic growth phases of the cells within a bioreactor, whereas favorable growth conditions during the lag and exponential growth phase may lead to the cells reaching the transition phase earlier. Unfavorable conditions may have the opposite effect, leading to a delayed transition phase. Based on the characteristic growth phase the cells are currently in, their response behavior to an applied factor setting differs, e.g., for cells already in the transition phase, a downshift in the temperature may be optimal to maintain high cell densities, whereas for cells still in the exponential growth phase, higher temperatures might improve process performance. This is especially relevant in a mammalian fed-batch process with cells being highly sensitive to changing environmental conditions [17].

To consider the potential effects of process history in the analyses, either the initial value of each stage or the previous value is used as input to the OLS- or hybrid model, respectively [10]. Interpretation of the influence of process history in semi-parametric models is hard. The process history’s impact on cellular response behavior is hidden in the weights of the neural network [13]. OLS is an easier to understand approach since the impact of the memory effect is expressed as the interactions of the stage-dependent factors with the initial value per stage [10]. The approach relies on the initial values being representative of the potential cellular states that may result from different process histories. The fulfillment of this assumption is not guaranteed, which makes evaluating the memory effect difficult. In the so-far used time series format for iDoE, the memory effect was not explicitly considered during planning. For a sufficiently powered and resolving iDoE using time series (OFOC-format), i.e., a design containing many combinations of factor settings and replicates of previous stage combinations being matched with relevant factor combinations for later stages, divergent initial values required to model memory effects might be obtained in the data. Balancing factor levels within bioreactors and stages is a measure to approach this need but may require performing additional experiments to reach adequate balance [10]. Analogous to interactions between two factors only being resolvable if both were varied jointly, a current factor setting needs to be systematically combined with different cellular states for the design to be able to directly model interactions between a current factor and process history. Since cellular states are a measured output and not directly controllable, multiple combinations of previous factors can be used to eventually generate different states, while some combinations might not elicit differences. The assumption of representative initial values is likely not fulfilled if the focus during planning was on reducing the number of bioreactors.

As a solution to the issue of describing process history during iDoE, we propose a new planning approach. The one-factor-multiple-columns (OFMC)-format explicitly describes the effect of each stage-dependent factor via separate columns. This format allows for the explicit consideration of across-stage interactions (ASIs) during planning. These describe the impact of previous stages on the current factor settings. For illustration purposes, we planned an iDoE oriented on the characteristic growth phases during a fed-batch process using the OFMC-format. One benefit was that OFMC-iDoE can directly be obtained from published design-generating algorithms without the need for manually adjusting the balance of factor levels. The experimental data were analyzed with two OLS approaches: stage-wise modeling based on the derived OFOC-format (OFOC-SW) is compared to direct day 14 modeling using the newly proposed OFMC-format (OFMC-D14). Since each row of the OFMC-iDoE contains all information, direct modeling of day 14 becomes feasible. Special emphasis is put on the consideration of process history and the impact of the memory effect. Researchers will be provided with a framework on how to plan iDoE and model the obtained data using the OFMC- and previously used OFOC-format. It will be shown how OFMC simplifies modeling and interpretation of the impact of the memory effect while supporting the intensification of performance via stage-wise optimized bioprocess dynamics.

## 2. Materials and Methods

### 2.1. Design Planning and Evaluation

An iDoE with 4 stages was planned. Stage 0 comprised the first 2 days of the process, where temperature (T) and dissolved oxygen saturation (DO) were at center level. Stage 0 was used as the initial adaption time of the cells coming from the N − 1 stage. Stages 1 (day 2–6), 2 (day 6–10), and 3 (day 10–14) lasted for 4 days each. Within each iDoE stage, the factor setting of T and DO could be varied, resulting in six numerical factors T1, T2, T3, DO1, DO2, and DO3. Each factor was investigated at 3 levels. A 48-run I-optimal response surface design (considering all main effects, 2-factor interactions, and quadratic effects) was planned using Design Expert^®^ Version 13 (StatEase, Minneapolis, MN, USA). Replicate runs were included to enable estimation of the pure error. The design was evaluated regarding power at a signal-to-noise ratio of 2 and 3 at a significance level of 5%. Correlation between model terms, a low prediction variance across 80% of the fraction design space, as well as the leverage of individual experimental runs were considered in the design evaluation.

### 2.2. Cell Culture

A CHO-K1 GS LPL knockout cell line producing an immunoglobulin G1 (IgG1) monoclonal antibody was cultivated in suspension using chemically defined media and feeds. Media and feeds are proprietary (Boehringer Ingelheim Pharma GmbH and Co. KG, Ingelheim, Germany). Seed cultures were performed in shake flasks and bag bioreactors until the N − 2 stage, followed by a controlled N − 1 batch culture carried out in a 3 L- glass bioreactor. The iDoE experiments were conducted in an automated microscale bioreactor (ambr^®^250™) system (Sartorius Stedim Biotech GmbH, Goettingen, Germany) and were operated in fed-batch mode throughout the entire process of 14 days. All bioreactors were inoculated from the same 3 L pre-stage. T, DO, and pH were controlled inline using sensors. T and DO setpoints were changed according to the planned iDoE. Starting from day 2, the nutrient feed containing glucose and the 2nd feed were added with a constant rate based on the starting volume of 3% (*v*/*v*) and 1% (*v*/*v*) per day, respectively. If needed to maintain an optimal concentration, glucose was added as a bolus on a day-to-day basis. Antifoam boli were automatically administered via the integrated foam sensor.

### 2.3. Process Analytics

Routine cultivation samples were taken at least once daily over the entire culture duration. Sampling was performed in the morning. An additional afternoon sample was taken on days 2, 4–8, and 10–13 of the cultivation. Total cell density (TCD) and viable cell density (VCD) were measured using an automated cell counter (Vi-CELL BLU, Beckman Coulter Inc., Brea, CA, USA) based on an image classification following trypan blue exclusion staining. Viability was calculated as a fraction of both. Glucose and lactate analyses were performed in cell free samples by photometric assays using an automated wet chemical analyzer (BioMajesty^®^ JCA-BM6010/C; DiaSys Diagnostic Systems GmbH, Holzheim, Germany). Offline pH, pCO_2_, and pO_2_ were determined with a blood gas analyzer (Rapidlab™, Siemens Healthcare GmbH, Erlangen, Germany). Titer was quantified with a Protein-A-HPLC method (ACQUITY Arc LC system; Waters™ Corporation, Milford, CT, USA).

### 2.4. Data Pre-Processing

All responses were scaled between 0 and 1 based on the observed respective maximum and minimum values for information protection reasons. A scaled response of 0 represents the smallest observed value for a respective response in the data set, whereas 1 represents the largest observed value. Factor settings were coded between −1 and +1. This coding is commonly applied to reduce the correlation between the factors and thereby improve model selection [18] (p. 24). Data pre-processing, visualization, and modeling was performed in JMP18 (SAS Institute, Cary, NC, USA).

### 2.5. Ordinary Least Squares Modeling

The data were modeled using polynomial OLS regression considering all main effects, two-factor interactions, quadratic effects, and a subset of three-factor interactions depending on the used design. The assumed underlying statistical polynomial population model can be described as follows:(1)$$y_{l}=\beta_{0}+\sum_{i=1}^{8}\left(\beta_{i}x_{i}+\beta_{ii}x_{i}^{2}\right)+\sum_{i=1}^{8}\sum_{j<i}\beta_{ij}x_{i}x_{j}+\sum_{i=1}^{8}\sum_{j\neq i}{\beta_{iij}x}_{i}^{2}x_{j}+\sum_{i=1}^{8}\sum_{j<i}\sum_{k<j}\beta_{ijk}x_{i}x_{j}x_{k}+\varepsilon_{l}$$
where:


$y_{l}$ = lth observed response (e.g., viable cell density)$\beta_{0}$ = model intercept$\beta_{i}$ = i^th^ main effect$\beta_{\mathrm{ij}}$ = ij^th^ two-factor interaction$\beta_{\mathrm{ii}}$ = ii^th^ quadratic effect$\beta_{\mathrm{iij}}$ = iij^th^ partial cubic interaction$\beta_{\mathrm{ijk}}$ = ijk^th^ three-factor interactionx = $\left[T1,T2,T3,\mathrm{DO}1,\mathrm{DO}2,\mathrm{DO}3,\mathrm{Runtime},\mathrm{Init}\right]$$\varepsilon_{l}$ = residual error, assuming ε ∼ N(0, $\delta^{2})$. δ is estimated based on the root mean squared error (RMSE) of each regression model


The estimated polynomial regression model is described as follows:(2)$${\hat{y}}_{l}={\hat{\beta }}_{0}+\sum_{i=1}^{8}\left({\hat{\beta }}_{i}x_{i}+{\hat{\beta }}_{\mathrm{ii}}x_{i}^{2}\right)+\sum_{i=1}^{8}\sum_{j<i}{\hat{\beta }}_{\mathrm{ij}}x_{i}x_{j}+\sum_{i=1}^{8}\sum_{j\neq i}{{\hat{\beta }}_{\mathrm{iij}}x}_{i}^{2}x_{j}+\sum_{i=1}^{8}\sum_{j<i}\sum_{k<j}{\hat{\beta }}_{ijk}x_{i}x_{j}x_{k}$$
where:


${\hat{y}}_{l}(T1,T2,T3,D1,D2,D3,\mathrm{Runtime},\mathrm{Init})$ = l^th^ predicted response by the polynomial regression model (e.g., viable cell density)${\hat{\beta }}_{0}$ = estimate of model intercept${\hat{\beta }}_{i}$ = estimate of i^th^ main effect${\hat{\beta }}_{\mathrm{ii}}$ = estimate of ii^th^ quadratic effect${\hat{\beta }}_{\mathrm{ij}}$ = estimate of ij^th^ two-factor interaction effect${\hat{\beta }}_{\mathrm{iij}}$ = estimate of ii^th^ quadratic effect${\hat{\beta }}_{\mathrm{ijk}}$ = estimate of ijk^th^ three-factor interaction


Leveraging the OFMC-format, direct modeling of the day 14 time point data using Equation (1) with the eligible model terms provided in Table 1a was performed. A response surface model complexity was assumed including all model terms up to the second order of T1–T3 and DO1–DO3. Interactions occurring between factors of different stages are termed ASIs. Model selection departing from the baseline model was performed using pruned forward selection as implemented in JMP Pro^®^ [19] (pp. 318–319). Pruned forward selection estimates parameters using a combination of forward and backward steps, iterating a maximum of five times the amount of input parameters. Heredity was enforced, meaning that higher order terms can only enter if the lower order terms are already part of the model. Maintaining hierarchy is considered good practice when performing polynomial multivariate regression for predictive purposes [20]. The final model is selected based on the Akaike information criterion corrected for small sample sizes (AICc) [19] (pp. 624–625). The AICc should be used over AIC unless the ratio of runs to fitted parameters is greater than 40, which is not the case in our study [21]. Model selection based on AICc finds important predictors while penalizing overly complex models. Individual model validity in terms of OLS modeling assumptions was evaluated based on visual inspection of normal distribution and homoscedasticity (constant variance) of the residuals [10].

Stage-wise modeling of the time segment data using the OFOC-format was performed similar to [10]. Previously, the initial VCD value at the beginning of a stage was used to approximate the influence of process history. Respective values were determined for each bioreactor and considered as an additional factor (Init-VCD) in the polynomial regression model. The following adaptations were made: No stage-wise rebasing of the data was performed. Instead of using the Init-VCD per stage as a common input factor for all responses, response-specific Init factors were used. The usage of response-specific Init factors was previously suggested by [10]. Init values were determined for stages 2 and 3 of the iDoE. For stage 1, no Init factor was needed since reactors had equivalent process histories during days 0–2. No titer measurements were available before day 11. To still predict the Init titer value in stage 3, a model considering the main, quadratic, and intra-stage interaction effects of T1, T2, DO1, and DO2 was fitted. For each stage-wise model, a response surface complexity including three-factor interactions and partial cubic terms with runtime was assumed. The additional model complexity is required to describe complex temporal response behavior. All stage-specific model terms including runtime and the response-specific Init factors were considered. An overview of the eligible terms in the baseline models according to Equation (1) prior to selection is provided in Table 1b–d. Model selection and validation of OLS assumptions was performed as described for the OFMC-D14 modeling. To obtain a prediction for the final day 14 of the process, models were concatenated using the prediction of the previous stage as the Init value for the subsequent stage.

### 2.6. Model Evaluation

Individual OLS regression models were evaluated based on the coefficient of determination $\left(R^{2}\right)$, $R_{\mathrm{adj}}^{2}$, root mean squared error (RMSE_Ordinary_), and the predicted residual error sum of squares (PRESS) statistic $R_{\mathrm{pred}}^{2}$, RMSE_PRESS_ [22,23,24,25], and the F-Value. Furthermore, parametric 95% prediction intervals (PIs) were calculated based on the t-distribution. To evaluate differences in model predictions at day 14, 95% PIs were compared. OFOC-SW and OFMC-D14 model predictions were compared based on the root average squared error (RASE) and $R^{2}$.(3)$$\mathrm{RASE}=\sqrt{\frac{\sum_{l=1}^{n}{{(y}_{l}-{\hat{y}}_{l})}^{2}}{n}}$$(4)$$R^{2}=1-\frac{\sum_{l=1}^{n}{\left(y_{l}-{\hat{y}}_{l}\right)}^{2}}{\sum_{l=1}^{n}{\left(y_{l}-\bar{y}\right)}^{2}}$$
where:


n = number of observations$\bar{y}(T1,T2,T3,D1,D2,D3,\mathrm{Runtime},\mathrm{Init})$ = mean predicted response by the polynomial regression model over all ${\hat{y}}_{l}$


For each estimate post-selection in the OFMC-D14 models, the mean absolute effect estimate was calculated. A mean absolute effect estimate different from 0 means that the estimate was selected in at least one of the response models. The selected estimates in each response model are shown in Appendix A. The average absolute effect size was used to estimate the model term importance across response models.(5)$${\bar{\hat{\beta }}}_{i}=\frac{\sum_{m=1}^{4}\left|{\hat{\beta }}_{i}\right|}{4},\;{\bar{\hat{\beta }}}_{\mathrm{ii}}=\frac{\sum_{m=1}^{4}\left|{\hat{\beta }}_{ii}\right|}{4},\;{\bar{\hat{\beta }}}_{\mathrm{ij}}=\frac{\sum_{m=1}^{4}\left|{\hat{\beta }}_{ij}\right|}{4}$$
where:


${\bar{\hat{\beta }}}_{i}$, ${\bar{\hat{\beta }}}_{\mathrm{ii}}$, ${\bar{\hat{\beta }}}_{\mathrm{ij}}$= mean absolute estimate of i^th^ main effect, ii^th^ quadratic effect, ij^th^ two-factor interaction


Variable importance (VI) combining main and interaction effects in the total effect of a respective factor was evaluated using the access variable importance feature of JMP Pro^®^. Independent uniform inputs were used as recommended for designed experiments with nearly uncorrelated factors spread over the range represented in the study [26,27] (pp. 70–71).

Model optimization was performed based on the desirability function as implemented in JMP Pro^®^. The desirability function is a smooth piecewise function that passes through three user-defined control points (low, middle, high). The maximization function used consists of interpolating cubic polynomials between the control points and exponential functions in the tails [27] (pp. 64–65). Due to the exponential approximation, neither the low nor the high control point may be set to zero or one, respectively. Since all responses are on the same scale (0–1), the shape-controlling points were defined equally for all models. The low point desirability was set to 0.0660 for a scaled response value of 0.05. For a scaled response value of 0.5, the desirability was set to 0.5000. Large scaled response values of 0.95 were set to a desirability of 0.9819. The same desirability function was used for all three modeling approaches. An additional constraint of fixing culture day to 14 was applied for the OFOC-SW, time-series-based models. For maximizing the desirability, a gradient descent algorithm is used. Since no boundaries are set, the entire investigated factor range was eligible. To minimize the risk of finding a local optimum, multiple random starts were performed [27] (p. 65).

## 3. Results

### 3.1. Across-Stage Interactions Explicitly Account for Memory Effects

We developed an iDoE-format that explicitly considers and enables the quantification of potential memory effects via ASIs. The ASIs describe what influence a factor setting in a certain stage has on the effect of a factor setting in another stage on a measured response within one process run. The defined iDoE stages have been oriented on the characteristic phases of a mammalian fed-batch process. Stage division has been oriented on a pre-cultivation investigating cellular growth behavior at standard operating conditions (T = DO = 0). The growth curves are shown in Appendix A. Stage 0 addresses the initial lag phase stretching from day 0–2, where T and DO were kept at center level to allow adaptation to the N-stage. Stage 1 covered the exponential growth phase (day 2–6), stage 2 the transition phase (day 6–10), and stage 3 the production phase (day 10–14). The stage-dependent effect of T and DO during stages 1–3 has been explicitly considered, including all their interactions up to second order (Table 1a). To minimize variance across the design space for robust predictions, an I-optimal design was planned. Additionally, the correlation structure among model terms as well as the power to detect effects was investigated. The highest correlation was between DO2:DO2 and DO1:DO1 with −0.42, while correlations among all other model terms, excluding the intercept, were below |0.33|. The full correlation matrix is shown in Appendix A. The power to detect effects was high at an effect size relative to random error (signal-to-noise, SN) of 2, with the lowest being 0.96 for the interaction T3:DO1. The power at SN = 2 and 3, and the prediction variance at 80% coverage according to the fraction of design space plot are shown in Appendix A. The factor settings for T and DO during each stage are highlighted in the design matrix in Table 2. During each stage, all factor combinations within the design space were covered (Appendix A).

Based on the low correlation structure and high power, the design was deemed suitable to detect potential ASIs of stage-dependent factor effects. The question arises during which characteristic growth phase the effects of T and DO are most important and what the impact of ASIs is on process performance curves.

### 3.2. Impact of Stage 1 Factor Settings Dominates over Effects in Other Stages

The dynamics introduced by performing stage-wise factor setting shifts is shown for the VCD curves. Stage-dependent factor settings are visualized by colorizing the response curves (Figure 1). Based on the setting of T1, a division of the VCD curves into three groups can be observed (Figure 1a). Bioreactors run at a high T1 reach the highest peak VCDs, whereas bioreactors at a low T1 reach the lowest. Low values of T1 compared to high values of T1 delay VCD peaking into stage 3 instead of stage 2. The day 14 VCD values do not reflect the initial grouping based on T1 anymore due to the setting changes occurring during the process. Additionally, the level of DO1 had an impact on peak VCD, with bioreactors operated at low DO1 reaching the highest peak VCD and bioreactors operated at high DO1 resulting in the lowest peak VCD within the T1-dependent groups (Figure 1b). In addition to the dominant effects of stage 1 settings, the ASI T1:T2 showed prominent effects (Figure 1d). If T1 and T2 were equal and either at the +1 or −1-level, generally lower terminal VCDs were observed, whereas opposite settings of T1 and T2 within a bioreactor lead to higher terminal VCDs (Figure 1d). When looking at the T1-based grouping, a differential effect of T3 is apparent: bioreactors starting with a high or center-level T1 reach a higher VCD at day 14 if combined with a low T3 (Figure 1c). For bioreactors operated at a low T1, the effect of T3 is less clear from a visual inspection. The dominant effect of T1 is also present in the other investigated responses, with a high T1 leading to the highest terminal IVCD and titer but lowest terminal viability values as can be seen in Appendix A.

Summarizing the data description, stage-dependent factor settings have a significant impact on the performance response curves. Most prominently, T during the exponential growth phase led to a division into three subgroups. Moreover, the (memory) effect of the ASI T1:T2 was visualized. To investigate how the memory effect is captured during modeling, the previously suggested OFOC-SW analysis was compared to the here-proposed OFMC-D14 approach.

### 3.3. Memory Effect Has Significant Impact on Process Performance

While OFOC-SW only indirectly considers the memory effect through the Init factor, OFMC-D14 allows its explicit quantification through ASIs. Both modeling approaches are summarized in Section 2. A visual overview of the OFOC-SW approach as well as individual stage-wise models, fit statistics, and diagnostic residual plots can be seen for each response in the Appendix A. Slight heterogeneity was observed in some stage-wise models but not evaluated to be critical. This does not affect the coefficient estimates but can bias the estimates’ standard errors, which influences uncertainty intervals and model selection [28]. When assessing the predictive performance of the models, a certain inconvenience of the OFOC-SW models was apparent. To yield the full time series model, the stage models need to be concatenated. The time series was well predicted with the lowest R^2^ in the viability model and the highest RASE in the titer model (VCD: R^2^ = 0.9396, RASE = 0.0546; viability: R^2^ = 0.8720, RASE = 0.0775, IVCD: R^2^ = 0.9811, RASE = 0.0371, titer: R^2^ = 0.9147, RASE = 0.0817). Predictions of the concatenated models for all time points and bioreactors compared to the actual measurements are summarized in Appendix A. Typically, day 14 measurements of titer and viability are relevant for judging the suitability of a bioprocess, as this time point marks the harvest day and is used for optimization. Since the process history in stage-wise models is captured by the measured start value of each stage (Init), predictions for the endpoint of each stage-model need to be made and are used as the Init value for the subsequent stage models. Thus, making predictions for day 14 with stage-wise models is laborious and involves the nesting and concatenation of predictions from multiple models per response. The predictions for day 14 of OFOC-SW-based models can be obtained by fixing the culture day at 14 for the concatenated model. Since the effect of process history manifests through the Init factor predicted by the previous stages model, the stage 3 model with the actual Init value was additionally investigated. The differences in the prediction of the concatenated and stage 3 model come from the difference between the predicted and the actual Init value.

Using the herein-proposed OFMC-format, day 14 data can be directly modeled without loss of information since the settings of previous stages are recorded in the respective columns for the stages. Fit statistics, final models post-selection, and respective residual plots can be seen in the Appendix A. The actual by predicted plots for all responses for day 14 of the OFOC-SW-based stage 3 and the concatenated models are compared with the predictions of the direct timepoint model based on OFMC-D14 in Figure 2a–d. The performances of the OFMC-D14 and the OFOC-stage 3 models are comparable, while the concatenated OFOC-SW models performed worse, i.e., their R^2^ was lower and the RASE larger. In the viability model, the largest difference based on R^2^ and RASE can be seen between the concatenated OFOC-SW (R^2^ = 0.2287, RASE = 0.1751), OFOC-stage 3 (R^2^ = 0.7207, RASE = 0.1054), and OFMC-D14 (R^2^ = 0.7429, RASE = 0.1011) models. Overall, models for day 14 of viability are the least predictive, as can be seen, based on the lower R^2^ and higher RASE compared to the other responses (Table 3).

Taken together, the comparable performance of the OFMC-D14 and OFOC-stage 3 models suggests that both can predict day 14 for the investigated process performance responses. Differences between the stage 3 and concatenated OFOC-SW models come from differences in the predicted Init to the actual Init in stage 3. Model concatenation is required though to estimate the stage-dependent factor effects of preceding stages on day 14. The consideration of the memory effect using both modeling approaches and the quantification of the memory effects’ impact using OFMC-D14 are discussed in detail in the following. 

The effect of the process history in the OFOC-SW-models is captured by the Init factor. The influence of previous stages’ factor settings manifests in the model predicting the consecutive stages’ Init factor. In each stage 2 and stage 3 model, the Init factor has the largest effect estimate and appears often in interaction with the controlled factors. Interactions of the Init factor suggest that a dependence of the factor effects on the process history exists. The opposite effects of T2 on VCD depending on the Init factor settings with constant DO2 are shown in Figure 3a. When Init2_VCD is low, a high level of T2 leads to an increase in VCD at the end of stage 2. Conversely, when Init2_VCD is high, a high level of T2 has a negative impact and leads to a lower final VCD. Stage-dependent factor effects therefore must always be interpreted in combination with the respective Init factor settings. The estimates for the Init factor and its interactions in the stage-wise models for each response can be seen in Appendix A.

In contrast to the implicit consideration of process history through the Init factors in the OFOC-SW-models, the OFMC-format explicitly considers process history by including ASIs. Post model selection, 18 out of 27 model terms were selected in at least one investigated OFMC-D14 model (Appendix A). To assess the importance of the ASIs across responses, their magnitude relative to other selected terms was evaluated. For each model term estimate (${\hat{\beta }}_{i}$, ${\hat{\beta }}_{ii}$, ${\hat{\beta }}_{ij}$), the mean absolute estimate across all models was calculated according to Equation (5) (${\bar{\hat{\beta }}}_{i}$, ${\bar{\hat{\beta }}}_{ii}$, ${\bar{\hat{\beta }}}_{ij}$). A mean absolute estimate different from 0 means that in at least one response model the term was selected. Out of the 18 selected model terms, 8 were ASIs. The most important ASI and third most important model term T1:T2 (according to the mean absolute estimate) was selected in all models (Figure 3b). The negative estimate of T1:T2 in all models suggests that opposite directions for T1 and T2, which result in a negative coefficient of the interaction term, lead to higher response values. This confirms the observation of the ASIs’ impact on VCD in Figure 1d. The next most important ASIs are T2:DO1, T1:T3, and T3:DO1, which are especially important in the viability model (rank 9, 10, and 11). T1:T3 was also selected in all other models, except for titer. T2:T3 and DO1:DO3 appeared as influential ASIs for titer.

In conclusion, we could show that by using the OFMC-format, the effect of the memory effect on performance is explicitly considered and can be quantified via ASIs using standard OLS. This confirmed T1:T2 as being the most important ASI as well as additional ASIs playing crucial roles for individual responses, especially viability. The OFOC-SW models implicitly consider the memory effect through prediction of the consecutive stages’ Init by the previous stages’ model. Stage-dependent factor effects are dependent on the Init factor and must therefore be interpreted in a combined fashion. The main purpose of setting up regression models during development is to find optimal factor settings leading to a desired response behavior. Therefore, it was investigated if both modeling approaches would lead to similar optimizations despite their different form of considering the memory effect.

### 3.4. Multivariate Optimization for Day 14 Results in Similar Suggestions of the Stage-Wise vs. Direct Endpoint Modeling

Optimization was performed using the OFMC-D14 and concatenated OFOC-SW approaches. Since ideal factor settings for each stage should be predicted, the concatenated OFOC-SW model was used instead of the OFOC-stage 3 model. It was investigated if both approaches lead to similar optimal factor settings and model predictions using identical desirability functions. For the interpretation of the optimization, the importance of the individual factors in the respective model needs to be considered, i.e., for a factor with large variable importance (VI), the optimization suggestion is more impactful than for less important factors. Settings are shown for each factor that is part of a model post-selection for a respective response. The optima and associated predictions using the concatenated OFOC-SW and OFMC-D14 approaches are shown in Figure 4. The VCD predictions under the optimal input settings for the individual models are similar between the OFMC-D14 (0.63 ± 0.09) and concatenated OFOC-SW (0.64 ± 0.11) model. For maximizing VCD, high levels for T1 but low levels for T3 were determined to be influential and beneficial. Having fixed T1 and T3, the settings of the other input factors are not associated with improved VCD, i.e., the predictions for any setting of the other input factors fall within the 95% Pis. For IVCD, maximization for both models leads to similar input factor level suggestions with T1 being identified as the most important factor with a positive impact. For titer, again, high levels for T1 are most important for maximization. For DO1, which is the second most important factor for titer, both optimizations suggest smaller values. Model predictions of both approaches lay within the 95% Pis of each other. The biggest differences can be seen in the viability models. Proposed optimal factor settings for T1, T2, and DO1 are similar but opposite for T3, which is an important factor in both modeling approaches. For T1, T2, and DO1, low levels are proposed to be optimal using both modeling approaches. For T3, the OFOC-SW model proposes high levels, whereas the OFMC-D14 model proposes low level settings. Moreover, the concatenated OFOC-SW model evaluates T1 as far less important than the OFMC-D14 model (Figure 4). This is also the case for DO1, which is evaluated as less important in the OFOC-SW models compared to the OFMC-D14 models. Across responses, T1 is the most important factor. Generally, the stage-dependent T factors are more important than the stage-dependent DO factors. Whereas DO1 is among the most important predictors in the OFMC-D14 models, the impact of DO2 and DO3 on process performance is negligible in both modeling approaches. Nevertheless, they appeared to be important in some ASIs (Figure 2).

Optimization based on both modeling approaches leads to similar predictions and optimal factor setting suggestions for VCD, IVCD, and titer at day 14. Additionally, both modeling approaches identify T during the exponential growth phase as the most important factor. Given that the OFOC-SW viability model was identified as being less predictive for day 14, the differences in the optimization are likely due to the OFOC-SW approach failing to properly capture the influence of process history.

## 4. Discussion

In this work, a new format was introduced to plan and model data obtained from iDoE. The OFMC-format explicitly describes stage-dependent factor effects on the whole process, enabling the consideration of potential memory effects through ASIs. The impact of the stage-dependent factor effects and ASIs on the process performance responses VCD, IVCD, viability, and titer was visualized and statistically quantified. The simplified analysis leveraging the OFMC-iDoE was compared to previously published analysis approaches confined by the OFOC-format [10], especially highlighting the different forms of considering process history. Whereas OFOC-formats require laborious stage-wise time series modeling and can only indirectly consider process history through the introduced Init factor, the OFMC-format allows direct and thus simplified modeling of day 14. The performance of both modeling approaches to predict day 14 was compared, highlighting limitations of the OFOC-SW when concatenating individual stage-wise models. Furthermore, the impact of the memory effect on process optimization for day 14 was investigated using both modeling approaches. Predicted optimal factor settings and responses were similar for all responses except viability for which a significant impact of the memory effect was observed, which was not sufficiently captured by the OFOC-SW model. Considerations for planning and modeling iDoE data using OFOC-SW vs OFMC-D14 are summarized in Table 4 and will be outlined in the following.

### 4.1. Direct Description of Across-Stage Interactions Simplifies Modeling

By planning an iDoE using the OFMC-format, all information of input levels in previous stages is captured in the dedicated columns. Thus, a time column in the design is not mandatory, nor is the recording of time series. Consequently, the number of analytical samples needed for analysis can be reduced to sampling only at the time point of interest, e.g., day 14 as the end of process. Even though not necessary, time series modeling can still be performed using the OFMC-format. The modeling of time series using the OFOC-format comes with additional challenges due to the dependency of repeated measures in one bioreactor. This could be accommodated by using an altered covariance structure such as AR(1) but was not done in this work. Autocorrelation does not affect the mean prediction of the OLS model [29], which was of primary interest in this study, so we kept the analyses similar to the published approach [10]. The dependency of OFOC-SW on time series data limits its applicability since some responses, e.g., titer and product quality attributes, cannot be measured in the first days of the process. In fed-batch processes, usually, the cells start producing the product, in this case, the IgG1 antibody in the transition phase. Furthermore, multiple product quality measurements are generally undesired due to their labor intensity and high cost. To still be able to model titer in the OFOC-SW approach, time point modeling based on the OFMC-format can be used to describe the Init value, needed as input for the stage 3 model (Table 1b; Figure 2d).

The OFMC-format allows consideration of ASIs in the assumed underlying model structure for design generation. This explicit consideration of the memory effect via ASIs allows for its quantification. The interpretation of an ASI would be the impact of a specific stage-dependent factor on the response behavior of the cells to a factor setting of another stage-dependent factor (visualized for T1:T2, Figure 1). Analogously the impact of the ASIs is captured in OFOC-SW modeling through the Init factor. The in this work used OFMC-format can be easily converted to the OFOC-format. Vice versa, the conversion might not be possible due to aliasing of the ASIs with intra-stage model terms. This becomes more severe when run counts are reduced leading to increased correlation and potentially wrong attribution of observed effects to model terms. Especially when using iDoE as a tool to reduce the experimental burden as previously suggested, this should be carefully considered [8,11,14]. Planning the input factor settings in the different stages in a combined fashion and not separately enables a classical OLS modeling approach for day 14 data and thus simplifies modeling and interpretation: Data from OFMC-iDoE can be directly captured in one model per response and does not need to be split stage-wise into segments with different Init values that need to be modeled and evaluated individually prior to re-concatenation for final model predictions. Having one model instead of several for one response eases optimization as the temporal dependence of factor effects is inherent in the model and thus no reverse-engineered stage-wise optimization is needed. Moreover, previous data obtained from classical DoE investigating the same factors could be easily converted to the OFMC-format and used as a starting point making efficient use of prior knowledge. The convenience of simplified data acquisition and analysis using OFMC comes at the cost of an increased run number. This is due to the higher number of input factors and associated (across-stage) interactions and quadratic effects. This could potentially be addressed by including prior knowledge about the importance of ASIs during future iDoE planning. Unlikely (across-stage) interactions and quadratic terms could be excluded, performing alias optimal supersaturated designs [30]. If a reduction in the assumed underlying model complexity is biologically sound, orthogonal minimally aliased response surface (OMARS)-designs would also be applicable. OMARS-designs show great promise in resolving response surfaces with limited amounts of active factors [31] as has been the case in our data.

### 4.2. Analysis of Process Outcomes Promotes Robustness of the Analysis Against Shifted Process Kinetics

Our data show that different process histories can lead to a shift in the growth curves. For bioreactors run at T1 = −1, the transition phase stretches over the second and third stage, leading to a shorter production phase. For bioreactors run at T1 = 1 or 0, the VCD curves peaked in the second stage following the intended growth-stage division of the process. Bioreactors run at T1 = 1 reach the transition and production phases the earliest, leading to the longest production phase (Figure 1). Similar observations of shifts in characteristic growth phases have been made in previous studies where different time points and magnitudes of temperature shifts were investigated for CHO production cells [32]. The observed VCD peak times were less variable compared to this study though. These shifted kinetics cause a problem to OFOC-SW modeling since stages are fixed, and describing the time course of shifted response curves becomes increasingly difficult due to increased shape variability. The different shape of the response curve based on the previous stages’ factor settings needs to be described based on the interactions with the Init factor. The more stages were planned, the shorter their duration and thus the risk of a stage shift. Moreover, the risk increases of stage-dependent factor effects not manifesting within their respective stage due to too short adaption times, falsely attributing their effect to the consecutive stage. By implementing only three stages with a duration of four days each, which resemble the characteristic culture phases of fed-batch processes, the risk of an input combination intended to, e.g., study the transition phase occurring only in the production phase, is reduced. Moreover, a stage duration of four days is in line with reported adaptation times for CHO cells to mild factor shifts [15]. In the OFMC-format, stage-dependent factor effects and their interactions on the whole process are explicitly captured, which makes the associated analysis more robust against shifted kinetics than the OFOC-SW modeling.

### 4.3. Transient Importance of Input Factors Can Be Directly Assessed

Comparison of the VI of input factors by stage based on the OFMC-D14 models has shown that the factor settings during the exponential growth phase are most important (Figure 3). This holds true across all investigated responses and could be due to the fact that factors of the first stage have more time for their influence to manifest through interactions with factors of later stages. Alternatively, the cells in culture undergo several divisions and are highly dynamic during the exponential growth phase, which could render them more sensitive to input factors. While T in later stages was associated with relatively large effects too, for DO, only a high importance in stage 1 relative to DO2 and DO3 was seen. Investigating the impact of DO in a time-invariant manner would have diluted the high impact of DO during the exponential growth phase, resulting in an underestimation of its impact. It has been suggested that high DO concentrations during the production phase could lead to oxidative stress being critical for cellular survival [10,33]. This was not the case in this study, which showed much stronger effects of DO during the exponential growth phase. While high temperatures during the exponential growth phase are beneficial to maximize VCD and titer, lower temperatures during the production phase maintain high VCDs and viability. The results of the optimization in Figure 4 suggest that an increase in T1 beyond the investigated range could have led to even higher titers and IVCDs, which would likely result in lower viabilities at day 14 though. Similarly, the regression models indicate that lower DO concentrations would have been more optimal to maximize viability at day 14. This would need to be investigated in further experiments though since regression models are not suited for extrapolation. The positive impact of a temperature downshift on CHO cultivations has been confirmed by many other studies [32,34,35,36]. The high amount of selected ASIs among the response models underlines their high overall importance (8/18 selected terms). Among the ASIs, T1:T2 is the most important, being selected in all models (Figure 3b). Its estimated coefficient is negative, suggesting that opposite directions of factor settings for T1 and T2, thus a shift during the process, is favorable for maximizing process performance. While a high T1 leads to a high VCD that should be maintained by limiting decreases in viability through low T2, a low T1 leads to a lower VCD which can partly be compensated by a high T2.

### 4.4. Interpretability and Applicability of Models Differ

The stage-wise modeling approach performs well when time series data are available to predict performance response curves (Appendix A, R^2^ ≥ 0.8720, RASE ≤ 0.0817). However, optimization of the process endpoints is often of bigger interest than the entire response trajectory. The additional effort of predicting all time points is needed though to obtain a prediction for the last time point with OFOC-SW concatenated models. Notably, the concatenated models perform worse compared to the stage 3-models for any response, while stage 3 and OFMC-D14 models’ performances are comparable (Figure 2, Table 3). The worse capability of the concatenated OFOC-SW model is expected due to the multiplicity of modeling and the propagation of the error from previous stages via the Init factor. Deviations in the predictions of earlier stages models translate to deviations in the predicted from the actual Init factor. The uncertainty in the prediction of the Init factors would also translate to increased uncertainty intervals during stages 2 and 3. This was not considered, instead using the uncertainty intervals of stage 3 corrected by the predicted Init values. The OFOC-stage 3 model would therefore be the best achievable prediction if the Init factor was predicted exactly by previous models. One reason for bad performance of the OFOC-SW model is when limited variability is observed during a stage. This is, for example, the case for the viability data during the first days. With the stage-wise modeling approach, almost no changes in the response are detected, despite the differences in the input factors (Appendix A, stage 1). Thus, the viability model could not capture the importance of T1 and DO1, while it is evident from the OFMC-D14 analysis, that T1 and DO1 have a strong (memory) effect on viability too. Their limited effect detection in the stage 1 model for viability is the reason for the poor performance of the concatenated stage-wise viability model for day 14 (Figure 2c; Table 3, R^2^ = 0.2287, RASE = 0.1751). Poor performance was also described in previous studies when modeling viability during the exponential growth phase due to limited variability [9]. Furthermore, ASIs are of high importance in the OFMC-D14 model of viability. The most important ASIs all involve T1 or DO1 as can be seen in Figure 3b, which were underestimated by the stage-wise model. If the models for previous stages are not predictive, the resulting Init factor values as proxy for process history and memory effects cannot be captured by the stage-wise models. The effect of the previous stages’ factor settings needs to directly manifest to be captured by the Init factor. As a result, different optimal factor settings were suggested with the concatenated OFOC-SW vs OFMC-D14 model (Figure 4). The viability example illustrates that when little variation is observed on any stage level, the stage-wise modeling approach has trouble correctly estimating factor importances and considering the impact of the memory effect, putatively resulting in false optimization.

## 5. Conclusions

Well-designed intra-experimental setpoint changes are the key element of iDoE. They allow for the characterization and optimization of characteristic growth phases during fed-batch processes. However, it is mandatory to properly consider the dependence of factor effects on the process history when planning and analyzing data from iDoE. These dynamics need to be adequately captured by the selected modeling approach. The herein-introduced OFMC-format to plan and model iDoE explicitly considers the impact of input settings from previous stages on the current stage via ASIs. Thereby, a straightforward quantification and interpretation of the influence of process history on the response behavior in a temporally resolved manner becomes feasible. Across the modeled responses, the factor settings during the exponential growth phase were consistently most important relative to the settings in later process stages. The assessment of effects per stage ensures that input factors with a timely limited impact, e.g., DO1 during the exponential growth phase, are not overlooked. In addition to its capacity to describe time-dependent effects, the proposed OFMC-format enables reduced sampling. Compared to the previously published OFOC-format, one timepoint instead of a whole time series with multiple collections per day is feasible. This is due to the explicit description of factor setting per stage in their own columns. Thereby, a straightforward OLS modeling approach for day 14 becomes feasible, removing the necessity for complicated stage-wise timeseries modeling and subsequent concatenation, resulting in improved end-of-process predictions of the OFMC-D14 model for each investigated response compared to the previously used format. The convenience of simplified data acquisition and analysis comes at the cost of an increased run number due to the higher number of input factors and associated second order effects. This article is meant as a framework to help experimenters plan iDoE to perform dynamic optimization of their processes. All necessary considerations concerning planning and modeling using the OFOC- or OFMC-format with emphasis on the influence of process history are provided.

To further position iDoE as a versatile tool in upstream processing, more efficient design formats should be investigated to reduce the experimental burden. Moreover, the influence of intra-experimental factor setpoint shifts on product quality needs to be elucidated. Finding and validating a sweet spot that increases process performance while controlling PQ in desirable ranges thanks to knowing about critical time periods would be a great step towards quality by (intensified) design (of experiments). Additionally, we see the potential for iDoE to generate more robust bioprocesses in line with cellular dynamics. Especially for complex molecule formats, screening for time windows when cells are susceptible to certain input factors could be a handy way to individually adjust existing platforms such that they ensure appropriate yields, quality, and process robustness.

## Figures and Tables

**Figure 1 bioengineering-12-00319-f001:**
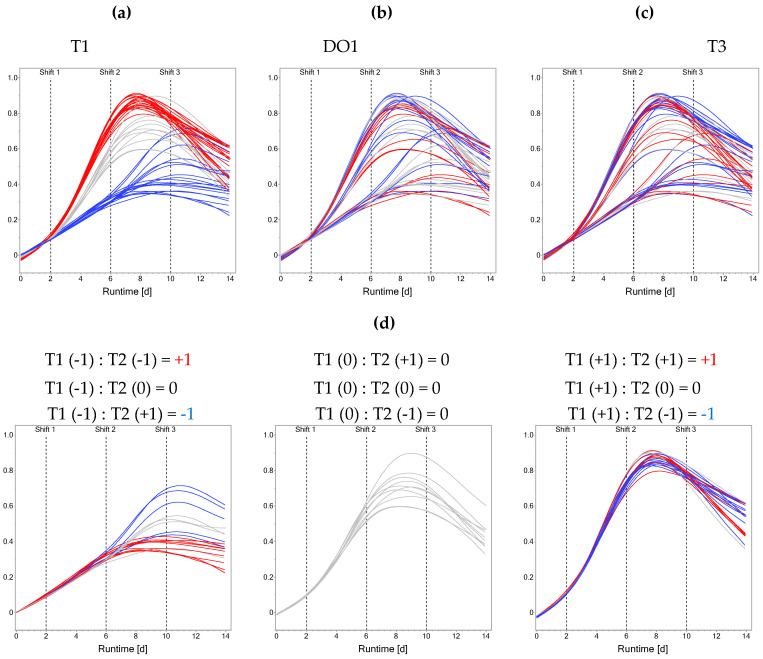
Influence of stage-dependent iDoE factor settings on viable cell density. Colorization is based on T1 (**a**), DO1 (**b**), T3 (**c**), and on the across-stage interaction T1:T2 (**d**). Red color indicates high level, grey indicates center level, blue indicates low level. Shift times of factor settings at days 2, 6, and 10 are indicated by dashed lines. Smoothing has been applied for visualization. For (**d**), bioreactors ran at T1 = +1, T1 = 0 and T1 = −1 have been separated and the factor combinations leading to different colorations are indicated.

**Figure 2 bioengineering-12-00319-f002:**
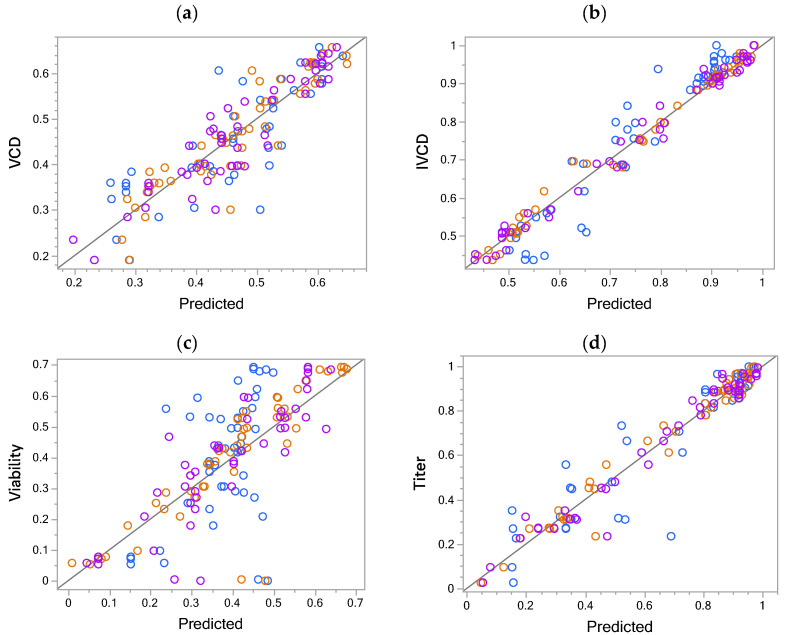
Predictive performance for models of day 14 data. Actual by predicted plots for (**a**) viable cell density, (**b**) integral viable cell density, (**c**) viability, and (**d**) titer of the day 14 of the concatenated one-factor-one-column stage-wise (OFOC-SW) (blue), stage 3 (orange), and one-factor-multiple-columns (OFMC-D14) models (purple).

**Figure 3 bioengineering-12-00319-f003:**
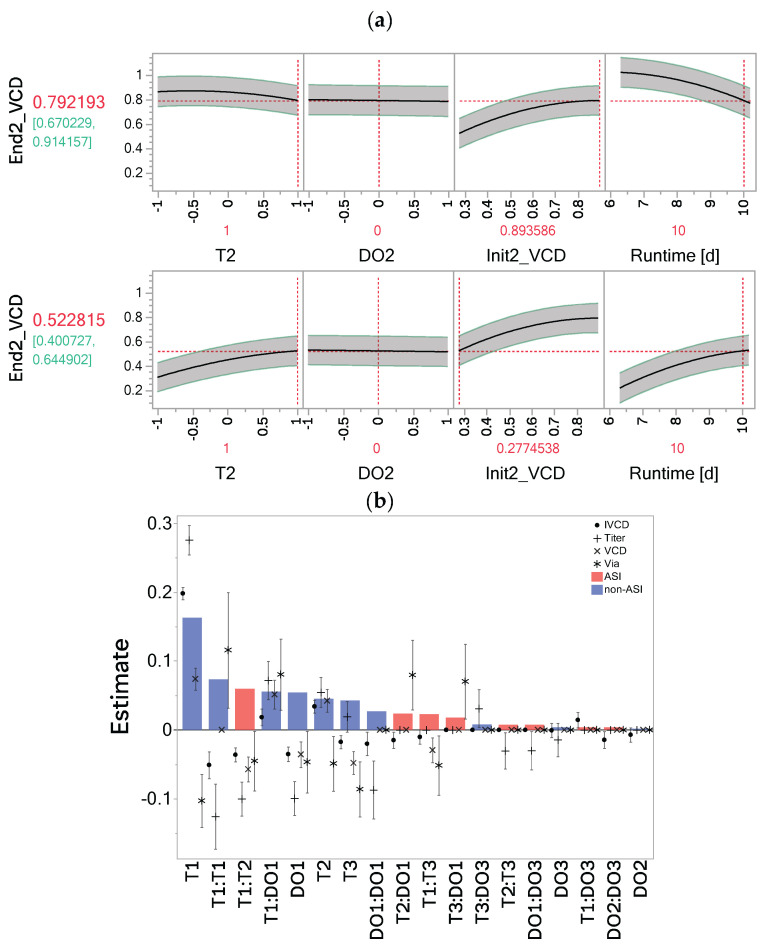
Impact of the memory effect in the one-factor-one-column stage-wise (OFOC-SW) and one-factor-multiple-columns day 14 models (OFMC-D14). (**a**) Opposite effects of T2 on viable cell density depending on the Init2_VCD level. (**b**) Effect size estimates ${\hat{\beta }}_{I}$, ${\hat{\beta }}_{ii}$, ${\hat{\beta }}_{ij}$ post model selection for the individual responses (viable cell density, integral viable cell density, viability, titer). Estimates are sorted by the overall importance of model terms based on the mean absolute estimates ${\bar{\hat{\beta }}}_{I}$, ${\bar{\hat{\beta }}}_{ii}$, ${\bar{\hat{\beta }}}_{ij}$ shown as bars (for mean absolute estimates different from 0). Across-stage interactions (ASIs) are displayed in red, non-across-stage interactions (non-ASIs) in blue. End2_VCD = terminal VCD value at the end of stage 2; Init2_VCD = initial value of VCD at stage 2. Dashed red lines indicate respective factor settings.

**Figure 4 bioengineering-12-00319-f004:**
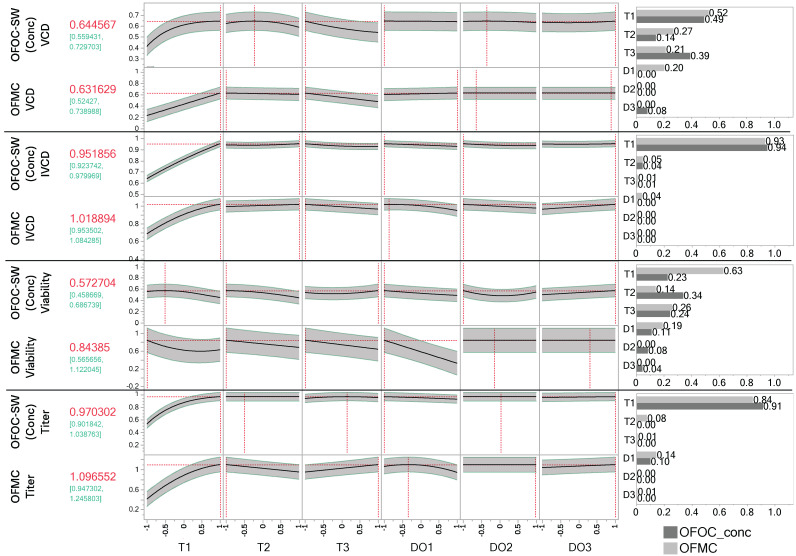
Comparison of model predictions and proposed optimal settings of the concatenated stage-wise one-factor-one-column (OFOC-SW) and one-factor-multiple-columns endpoint analysis for day 14 (OFMC-D14). Prediction profiler set to maximize the individual response (95% prediction interval in green). The results of the optimization for viable cell density (VCD), integral viable cell density (IVCD), viability, and titer including factor settings, and overall variable importance are shown. Dashed red lines indicate the optimal stage-dependent factor settings for a respective modeling approach.

**Table 1 bioengineering-12-00319-t001:** Eligible model terms of the polynomial regression model prior to model selection. (**a**) Model terms considered in the one-factor-multiple-columns format for modeling day 14 data (OFMC-D14). (**b**–**d**) Model terms considered in the one-factor-one-column stage-wise format (OFOC-SW) modeling each stage separately using time series data. (**e**) Model to predict the stage 3 Init for titer. T1,2,3 = temperature (day 2–6, 6–10, 10–14), DO1,2,3 = dissolved oxygen (day 2–6, 6–10, 10–14), Init = initial value of a response at the beginning of a stage.

(**a**) **OFMC-D14**		
T1, T2, T3, DO1, DO2, DO3, T1:T1, T1:T2, T2:T2, T1:T3, T2:T3, T3:T3, T1:DO1, T2:DO1, T3:DO1, DO1:DO1, T1:DO2, T2:DO2, T3:DO2, DO1:DO2, DO2:DO2, T1:DO3, T2:DO3, T3:DO3, DO1:DO3, DO2:DO3, DO3:DO3		
(**b**) **OFOC-SW: Stage 1**	(**c**) **OFOC-SW: Stage 2**	(**d**) **OFOC-SW: Stage 3**
T1, DO1, Runtime, T1:T1, T1:DO1, DO1:DO1, T1:Runtime, DO1:Runtime, Runtime:Runtime, T1:T1:Runtime, T1:DO1:Runtime, DO1:DO1:Runtime	T2, DO2, Init2, Runtime, T2:T2, T2:DO2, DO2:DO2, T2:Init2, DO2:Init2, Init2:Init2, T2:Runtime, DO2:Runtime, Init2:Runtime, Runtime:Runtime, T2:T2:Runtime, T2:DO2:Runtime, DO2:DO2:Runtime, T2:Init2:Runtime, DO2:Init2:Runtime, Init2:Init2:Runtime	T3, DO3, Init3, Runtime, T3:T3, T3:DO3, DO3:DO3, T3:Init3, DO3:Init3, Init3:Init3, T3:Runtime, DO3:Runtime, Init3:Runtime, Runtime:Runtime, T3:T3:Runtime, T3:DO3:Runtime, DO3:DO3:Runtime, T3:Init3:Runtime, DO3:Init3:Runtime, Init3:Init3:Runtime
(**e**) **Init titer for stage 3**		
T1, T2, DO1, DO2, T1:T1, T2:T2, T1:DO1, DO1:DO1, T2:DO2, DO2:DO2		

**Table 2 bioengineering-12-00319-t002:** Design matrix of the performed I-optimal design considering the full response surface model of the stage-dependent effects of temperature and dissolved oxygen.

	Stage 1 (Day 2–6)		Stage 2 (Day 6–10)		Stage 3 (Day 10–14)			Stage 1 (Day 2–6)		Stage 2 (Day 6–10)		Stage 3 (Day 10–14)	
Run	T1	DO1	DO2	T2	DO3	T3	Run	T1	DO1	DO2	T2	DO3	T3
1	−1	1	−1	−1	−1	−1	25	1	−1	−1	−1	1	0
2	1	0	0	−1	0	−1	26	0	−1	0	1	0	−1
3	1	−1	−1	1	−1	−1	27	1	0	0	0	1	−1
4	1	0	0	−1	0	−1	28	0	0	0	0	0	0
5	−1	1	−1	1	−1	1	29	1	−1	0	0	−1	1
6	−1	0	0	−1	0	−1	30	0	1	1	−1	1	−1
7	1	1	−1	1	1	−1	31	−1	0	1	0	0	1
8	1	1	−1	−1	1	1	32	0	1	1	1	1	1
9	−1	0	−1	−1	1	−1	33	0	0	0	−1	−1	0
10	1	1	1	−1	1	0	34	1	0	0	1	1	0
11	−1	1	1	1	1	0	35	−1	1	0	0	0	0
12	−1	−1	−1	−1	−1	1	36	1	1	−1	−1	−1	−1
13	−1	−1	−1	1	1	1	37	−1	−1	1	−1	0	0
14	1	0	0	1	0	1	38	1	−1	1	−1	−1	−1
15	−1	0	0	−1	0	−1	39	−1	0	−1	1	0	−1
16	1	0	0	1	0	1	40	0	1	−1	−1	0	0
17	−1	1	1	−1	−1	1	41	0	0	−1	0	0	0
18	−1	−1	1	1	−1	0	42	0	0	0	0	0	0
19	−1	0	0	−1	0	−1	43	0	−1	−1	0	0	1
20	1	−1	1	−1	1	1	44	0	−1	1	−1	−1	1
21	1	−1	1	1	1	−1	45	1	1	−1	−1	−1	1
22	1	1	1	1	−1	−1	46	−1	−1	0	0	1	−1
23	1	0	0	1	0	1	47	−1	0	0	−1	1	1
24	1	0	0	−1	0	−1	48	−1	0	1	0	−1	−1

Coloration indicates high (red), low (blue), and center level (white).

**Table 3 bioengineering-12-00319-t003:** Model predictions for day 14 of the concatenated (Conc), Stage 3 (S3) one-factor-one-column-stage-wise (OFOC-SW), and one-factor-multiple-columns timepoint models (OFMC-D14). RASE = root average squared error.

Model	Predictor	R^2^	RASE
OFOC-SW(Conc)	VCD	0.6755	0.0673
OFOC-SW(S3)	VCD	0.8433	0.0468
OFMC-D14	VCD	0.8588	0.0444
OFOC-SW(Conc)	IVCD	0.9020	0.0600
OFOC-SW(S3)	IVCD	0.9891	0.0200
OFMC-D14	IVCD	0.9867	0.0221
OFOC-SW(Conc)	Via	0.2287	0.1751
OFOC-SW(S3)	Via	0.7207	0.1054
OFMC-D14	Via	0.7429	0.1011
OFOC-SW(Conc)	Titer	0.8591	0.1111
OFOC-SW(S3)	Titer	0.9753	0.0465
OFMC-D14	Titer	0.9666	0.0541

**Table 4 bioengineering-12-00319-t004:** Comparison of the one-factor-multiple-columns stage-wise (OFOC-SW) modeling of time series and the one-factor-multiple-columns time point modeling (OFMC-D14) for iDoE data.

OFOC-SW	OFMC-D14
**Planning**	
Response surface model for each stage Manual correction/constraint to balance factor levels within bioreactor and stage Response variability on a stage level is needed	Response surface model with stage-dependent iDoE factors considered Many factors require a lot of runs Unlikely 2 factor interactions, quadratic effects that can be excluded?
**Execution**	
Time series data are required (multiple sampling per stage, before/after shifts) for each response	Routine sampling/endpoint sampling sufficient
**Modeling**	
Stage-wise OLS regression models with Init and culture duration as additional factors Concatenation of stages based on Init factor to yield whole-process model Errors in models of previous stages are propagated to the next stage (false Init value prediction) Time series modeling required, only D14 predictions are used for optimization	Classical OLS regression model (simultaneously considering stage-dependent factors and interactions impact on D14)
**Interpretation of factor effects**	
Impact of individual factors on a stage level should be interpreted in combination with the respective Init factor Impact of individual factors on the whole process can be estimated for the concatenated model	Explicit description of the influence of each stage-dependent factor, interaction, across-stage interaction, and quadratic effect on the whole process
**Consideration of the memory effect**	
Stage-dependent factors of earlier stage models influence consecutive stages’ models by their cumulative influence on the Init factor Effect of process history needs to manifest within the modeled stage Accurate prediction of the Init factor is crucial to predict stage-dependent factor effects	Explicit consideration during planning and quantification via across-stage interactions
**Quality checks**	
Standard statistical metrics (R^2^, $R_{pred}^{2}$, RMSE) Regression assumptions/investigation of residuals: linear, normally distributed, equal variances	

## Data Availability

Scaled raw data including OFOC-SW and OFMC-D14 analysis are included in the Appendix A (Sup_RawData.xlsx, Sup_one_factor_multiple_column(OFMC)_D14.jmp, Sup_one_factor_one_column(OFOC)_stage_wise(SW).jmp).

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
