# Peer review of "Direct Consideration of Process History During Intensified Design of Experiments Planning Eases Interpretation of Mammalian Cell Culture Dynamics"

_bioengineering, 2025, doi:10.3390/bioengineering12030319_

Round 1
Reviewer 1 Report
Comments and Suggestions for Authors
This manuscript dealt with using a new model, one-factor-multiple-columns (OFMC)-format, to explicitly describe the effect of each stage-dependent factor in the production phase and thus consider across-stage-interactions during planning. The authors cultured CHO cells in Ambr250 to execute the DoE experiment for temperature and dissolved oxygen in three stages. VCD, integral VCD, viability, and the titer were measured for the result. The manuscript is well written. I only have several suggestions which may improve the readability and the significance.
1. Figure 1 shows the overall factor level in each stage and tried to cover all the combinations. However, there are too many conditions to track or identify any trend. Moreover, each point on the x-axis is actually a period, and the between is a time point. And the continuity of the lines does not represent the real conditions. Probably, a table containing 48 conditions is easier to show the combinations of factors.
[T1, T2, T3] [DO1, DO2, DO3]
Condition 1 [+, 0, -] [+, 0, -]
Condition 2
2. Y-axis should represent VCD in Figure 2, but I am not sure the reason that VCD has to be normalized to 1. The same question for Figure 4 and 5.
Supplementary Materials contain some cell growth figures which may address this. However, I can not see supplementary figures on my end and this link (www.mdpi.com/xxx/s1) provided in Supplementary Materials on Page 20 did not show the content.
3. Figure 4a, the y-axis should be labeled as End2_VCD instead of VCD only, and can be more obviously separated from Init2_VCD in the 3rd column of the x-axis. Runtime2 here is from 0 to 4. I thought it should be from Day 6 to Day 10 for Stage 2.
4. Table 3 is the most important table to compare OFOC-SW and OFMC. Although the authors compared the differences side by side, the most important thing is to show that the new model can either have a higher prediction or require fewer runs. The potential risks should be added.
Author Response
Please refer to the attached file for the responses to the comments as well as the reviewed manuscript and supplements.

Reviewer 2 Report
Comments and Suggestions for Authors
The major issue that this manuscript claims that that the DoE planning eases the analysis of mammalian cell culture is not justified. The parameters used were encrypted or insufficient information is provided that it may not be able to benefit the readers or even for verifications by the scientific community.
The conclusion is poorly written.
Other comments are as follows:-
The title is not suitable with the words “Explicit Consideration” for a research paper. Do revise your paper title to reflect the actual work done.
“All responses were scaled between 0 and 1 based on the observed respective maximum and minimum values for information protection reasons. ” This defeats the purpose of academic publication, that other researchers can repeat your experiment and verify your results.
What is pp 70-71, pp64-65?? And p65?
What product do you refer to? “The dependency of OFOC-SW on time series data limits its applicability, since some responses, e.g., titer and product quality attributes, can not be measured in the first days of the process since sufficient amounts of the product need to be formed first. ”
The explanation of the memory effects is quite generalised.
Can you explicitly explain the factors and process history involved when culturing the Chinese hamster ovary cells?
Please check English in these sentences. “terms provided in Table 1, a was performed. ” “To still predict the Init titer value ”
What is the smallest sample size? “final model is selected based on the Akaike information criterion corrected for small sample sizes (AICc) .
In section 2.3, it is not clear of what cell density was applied.
The sampling interval should be more consistent instead of writing, “Additional sampling was performed on days 2, 4-8, 10-13 ”.
The conclusion is too long and needs to be more concisely written. Do not include discussions in the conclusion; include the main findings.
Author Response

(The authors gave the same response as above.)

Reviewer 3 Report
Comments and Suggestions for Authors
Review of the article "Explicit Consideration of Process History During Intensified Design of Experiments Planning Eases Analysis of Mammalian Cell Culture Dynamics"
The approach presented in the article may be considered controversial, since in process control usually the current state of cells is mainly taken into account.
My points to correct the article:
1. The aim of the work should be clearly stated in the Abstract, i.e., "The aim of this work is ..."
2. The article is weakly related to the metabolism pathways of cells used in the experiments.
More information about a CHO-K1 GS LPL knockout cell line producing an immunoglobulin G1 (IgG1) monoclonal antibody should be presented in the article along with metabolic pathways related to glucose metabolism.
3. It would be valuable to present by the Authors in the article the photo of the experimental setup along with its diagram.
4. In the Design Planning and Evaluation subsection is (line 127):
"...where temperature (T) and dissolved oxygen (DO) were at center level."
Please check it. Probably there should be "dissolved oxygen concentration (DOC)".
5. Optimal values of DO (or DOC), temperature, and pH for used cells should be presented. In the article, there should be added information on whether DO (or DOC), temperature, and pH were in the optimal ranges during the conducted experiments.
6. In the Cell Culture subsection is (lines 150-151):
"If needed to maintain an optimal concentration, glucose was added as a bolus on a day-to-day basis."
The way glucose is dosed is important and should be presented more clearly.
7. In the Abstract is (lines 16-19):
"Determining the influence of process history on the cellular responses, often referred to as memory effect, is fundamental for accurate predictions. However, the current iDoE designs do not explicitly consider nor quantify the influence of process history."
In given extracellular conditions, cellular response depends largely on the current intracellular state of the cells; for example, cellular response depends, among others, on the accumulation of intracellular metabolites, the accumulation of intracellular molecules, and the metabolic pathways that are active in a given moment.
From this point of view, determining the influence of process history on the cellular responses makes sense when the current intracellular state is mainly taken into account.
By examining process history, we can try to determine the intracellular state of cells and predict response in order to appropriately adapt the process control parameters. Without predicting the intracellular state of cells, it is rather impossible to predict the response of cells (on glucose in the considered case).
Please accordingly improve your article.
8. In the Introduction is (lines 78-80):
"Memory effects describe the influence that the process history has on cellular response behavior during later stages."
Memory effect can have influence, but this influence becomes less and less as we move further and further from the present moment. So, the most important are the current states of cells.
9. In the Introduction is (lines 81-83):
"This is especially relevant in mammalian fed-batch process with cells being highly sensitive to changing environmental conditions [17]."
This sentence is proof that continuous monitoring of the current states of cells is the most important, and decisions on changing control strategy should be made in accordance with the current states of cells.
Process history is, of course, important during the establishment of a control algorithm, but during the conduction of the process, decisions should be made taking into account mainly the current states of the cells.
10. In the Direct Description of Across-Stage-Interactions Simplifies Modeling is (lines 535-536):
"By planning an iDoE using the OFMC-format, all information of input levels in previous stages is captured in the dedicated columns."
In my opinion, in the future, as a continuation of this work, it would be valuable to consider the possibility of recognizing patterns formed by previous stages by an artificial neural network and modifying (or changing) the control parameters or control strategy in accordance with the recognized pattern.
Author Response

(The authors gave the same response as above.)

Round 2
Reviewer 3 Report
Comments and Suggestions for Authors
The Authors have appropriately handled my concerns. The article, after improvement, is better, and in my opinion, it can be published in Bioengineering.